# Prognostic accuracy of qSOFA score, SIRS criteria, and EWSs for in-hospital mortality among adult patients presenting with suspected infection to the emergency department (PASSEM) *Multicenter prospective external validation cohort study protocol*

**Abdullah M. Algarni**[1]*, **Musa S. Alfaifi**[2], **Ali A. Al Bshabshe**[3], **Othman M. Omair**[4], **Mohammed A. Alsultan**[5], **Hasan M. Alzahrani**[3], **Hadi E. Alali**[2], **Abdullah A. Alsabaani**[3], **Ali M. Alqarni**[6], **Salah A. Alghanem**[7], **Bandar S. Al Mufareh**[8], **Ayesha M. Almemari**[9], **Abdulrahman A. Sindi**[10], **Ibrahim U. Ozturan**[11], **Abdullah A. Alhadhira**[12], **Asaad S. Shujaa**[12], **Abdullah H. Alotaibi**[13], **Moosa M. Awladthani**[14], **Ahmed A. Alsaad**[15], **Abdullah A. Almarshed**[16], **AlHanouv M. AlQahtani**[17], **Tim R. Harris**[18], **Bader A. Alyahya**[19], **Saad A. Assiri**[20], **Feras H. Abuzeyad**[21], **Sara N. Kazim**[22], **Abdulrahman A. Al-Fares**[23], **Faisal Y. Almazroua**[24], **Naif T. Marzook**[25], **Abdullah A. Basri**[7], **Abdallah M. Elsafti**[18], **Abdulaziz S. Alalshaikh**[19], **Cansu A. Özturan**[26], **Yousef I. Alawad**[27], **Awad AlOmari**[28], **Malek A. Alkhateeb**[20], **Moonis M. Farooq**[21], **Latifa Ali AlMutairi**[29], **Muneera M. Alasfour**[29], **Mohammad I. Al Haber**[9], **Umma-Kulthum A. Umar**[9], **Nidal H. Bokhary**[10], **Saeed F. Alqahtani**[4], **Abdulrhman Almutairi**[24], **Hisham F. Alyahya**[24], **Wejdan S. Alzahrani**[25], **Fawziah Alsalmi**[25], **Abdulmajeed M. Omair**[3], **Faysal M. Alasmari**[3], **Sultan Y. Alfifi**[3], **Mohammed S. Al-Nujimi**[3], **Farid Foroutan**[30]

1 Family Medicine Department, Aseer Central Hospital, Abha, Saudi Arabia, 2 Emergency Medicine Department, Armed Forces Hospital Southern Region, Khamis Mushait, Saudi Arabia, 3 College of Medicine, King Khalid University, Abha, Saudi Arabia, 4 Emergency Medicine Department, Aseer Central Hospital, Abha, Saudi Arabia, 5 Saudi Commission for Health Specialties, Riyadh, Saudi Arabia, 6 Radiology Department, Prince Mashary Bin Saud Hospital, Belgraishi, Saudi Arabia, 7 Emergency Medicine Department, Bahrain Defence Force Hospital, Al Riffa, Bahrain, 8 Emergency Medicine Department, Royal Commission Hospital in Jubail, Jubail, Saudi Arabia, 9 Emergency Medicine Department, Shaikh Shakhbout Medical City, Abu Dhabi, United Arab Emirates, 10 College of Medicine, King Abdulaziz University, Jeddah, Saudi Arabia, 11 Kocaeli University, Faculty of Medicine, Emergency Medicine Department, Kocaeli, Turkey, 12 Emergency Medicine Department, Johns Hopkins Aramco Healthcare, Dhahran, Saudi Arabia, 13 Emergency Medicine Department, King Abdullah University Hospital, Riyadh, Saudi Arabia, 14 Critical Care Department, Armed Forces Hospital Oman, Muscat, Oman, 15 Emergency Medicine Department, King Fahad Specialist Hospital, Dammam, Saudi Arabia, 16 Emergency Medicine Department, King Fahad Medical City, Riyadh, Saudi Arabia, 17 Emergency Medicine Department, North Medical Tower Hospital, Arar, Saudi Arabia, 18 Emergency Medicine Department, Hamad Medical Corporation, Doha, Qatar, 19 College of Medicine, King Saud University, Riyadh, Saudi Arabia, 20 Emergency Medicine Department, Sulaiman Al Habib Medical Group, Riyadh, Saudi Arabia, 21 Emergency Medicine Department, King Hamad University Hospital, Muharraq, Bahrain, 22 Emergency Medicine Department, Rashid Hospital, Dubai, United Arab Emirates, 23 Critical Care Department, Al Amiri hospital, Kuwait City, Kuwait, 24 Emergency Medicine Department, King Saud Medical City, Riyadh, Saudi Arabia, 25 Emergency Medicine Department, King Fahad Armed Forces Hospital, Jeddah, Saudi Arabia, 26 Emergency Medicine Department, Gölcük Necati Çelik State Hospital, Gölcük, Kocaeli, Turkey, 27 Emergency Medicine Administration, King Fahad Medical City, Riyadh, Saudi Arabia, 28 Critical Care Department, Sulaiman Al Habib Medical Group, Riyadh, Saudi Arabia, 29 Emergency Medicine Department, Al Amiri Hospital, Kuwait City, Kuwait, 30 Health Research Methods, Evidence and Impact, McMaster University, Hamilton, Ontario, Canada

* abaid1406@hotmail.com, aalgarni4@moh.gov.sa, abaid1406@gmail.com



**Data Availability Statement:** No datasets were generated or analysed during the current study. All relevant data from this study will be made available upon study completion.

**Funding:** The author(s) received no specific funding for this work.

**Competing interests:** The authors have declared that no competing interests exist.

# Abstract

## Background

Early identification of a patient with infection who may develop sepsis is of utmost importance. Unfortunately, this remains elusive because no single clinical measure or test can reflect complex pathophysiological changes in patients with sepsis. However, multiple clinical and laboratory parameters indicate impending sepsis and organ dysfunction. Screening tools using these parameters can help identify the condition, such as SIRS, quick SOFA (qSOFA), National Early Warning Score (NEWS), or Modified Early Warning Score (MEWS). We aim to externally validate qSOFA, SIRS, and NEWS/NEWS2/MEWS for in-hospital mortality among adult patients with suspected infection who presenting to the emergency department.

## Methods and analysis

PASSEM study is an international prospective external validation cohort study. For 9 months, each participating center will recruit consecutive adult patients who visited the emergency departments with suspected infection and are planned for hospitalization. We will collect patients' demographics, vital signs measured in the triage, initial white blood cell count, and variables required to calculate Charlson Comorbidities Index; and follow patients for 90 days since their inclusion in the study. The primary outcome will be 30-days in-hospital mortality. The secondary outcome will be intensive care unit (ICU) admission, prolonged stay in the ICU (i.e., $\geq$72 hours), and 30- as well as 90-days all-cause mortality. The study started in December 2021 and planned to enroll 2851 patients to reach 200 in-hospital death. The sample size is adaptive and will be adjusted based on prespecified consecutive interim analyses.

## Discussion

PASSEM study will be the first international multicenter prospective cohort study that designated to externally validate qSOFA score, SIRS criteria, and EWSs for in-hospital mortality among adult patients with suspected infection presenting to the ED in the Middle East region.

## Study registration

The study is registered at ClinicalTrials.gov (NCT05172479).

## Introduction

Over the past decade, there has been continued focus on sepsis as a prevalent condition that accounts for 10% of admissions to intensive care units (ICUs) and is associated with a 10–20% in-hospital mortality rate [1–5]. Standardized protocols and physician awareness have significantly improved survival, but mortality rates remain high between 20% and 36%, with ~270,000 deaths annually in the United States [6–8]. It has been estimated that 80% of sepsis cases are identified and treated in the emergency department (ED), and the remainder develop sepsis during hospitalization with other conditions [7].

In 2016, the Society of Critical Care Medicine/European Society of Intensive Care Medicine (SCCM/ESICM) task force redefined sepsis based on organ dysfunction and mortality prediction [9–11]. Sepsis now is defined as life-threatening organ dysfunction caused by dysregulated host response to infection. This definition emphasizes the complexity of the disease that cannot be explained by infection or body response alone. Acute change in Sequential Organ Failure Assessment (SOFA) score ≥2 indicates sepsis-related organ dysfunction and is associated with in-hospital mortality. Systemic Inflammatory Response Syndrome (SIRS) and "severe sepsis" terms were omitted from the most recent definition. SIRS has been criticized for its poor specificity, while "severe sepsis" may underestimate sepsis's seriousness. A subset of patients may develop septic shock with underlying profound organ dysfunction and excess mortality. Clinically, septic shock is defined as persistent hypotension requiring vasopressors to maintain mean arterial pressure (MAP) ≥ 65 mm Hg and serum lactate level ≥ 2 mmol/L (18 mg/dL) despite adequate volume resuscitation.

Early identification of a patient with infection who may develop sepsis is of utmost importance [12]. Unfortunately, this remains elusive because no single clinical measure or test can reflect the complex pathophysiological changes in patients with sepsis. However, multiple clinical and laboratory parameters indicate impending sepsis and organ dysfunction. Screening tools using these parameters can help identify the condition, such as SIRS, quick SOFA (qSOFA), National Early Warning Score (NEWS), or Modified Early Warning Score (MEWS) (Tables 1 and 2) [13].

The 2016 SCCM/ESICM task force recommended using qSOFA [11], while the 2021 Surviving Sepsis Campaign strongly recommended against its use compared with SIRS, NEWS, or MEWS as a single screening tool for sepsis or septic shock [14].

Multiple studies have assessed qSOFA, SIRS, and EWSs validity in ED and showed conflicting results [15–21]. One systemic review compared qSOFA and EWSs (NEWS/Modified EWS [MEWS]) for predicting mortality and ICU admission when applied in the ED [13]. None of the eligible studies included NEWS2; and the authors of the review could not perform a meta-analysis due to marked heterogeneity in patient selection, definition of infection, outcomes, and settings. Moreover, studies have calculated the scores at different times. NEWS appeared more sensitive than qSOFA for predicting ICU admission and mortality at the commonly used thresholds (i.e., ≥2 for SIRS and qSOFA; ≥5 for NEWS, NEWS2, and MEWS), whereas qSOFA was more specific [13]. This correlates with previous criticisms of qSOFA, which have low sensitivity for early risk assessment [18–21].

We hypothesized that qSOFA has greater prognostic accuracy than SIRS and EWSs (NEWS/NEWS2/MEWS), and subsequently, aimed to reject the null hypothesis that all these

**Table 1. Component of qSOFA score and SIRS criteria.**

| Variable | qSOFA | | SIRS | |
|---|---|---|---|---|
| | *Cut-off* | *Points* | *Cut-off* | *Points* |
| Altered mental status (GCS <15) | Yes | 1 | — | — |
| Heart rate (beats/min) | — | — | >90 | 1 |
| Respiratory rate (breaths/min) | ≥22 | 1 | >20 | 1 |
| Systolic blood pressure (mm Hg) | ≤100 | 1 | — | — |
| Temperature (˚C) | — | — | <36 or >38 | 1 |
| White blood cells count (x10$^9$/μL) | — | — | <4 or >12 or >10% bands | 1 |
| | **Maximum score** | **3** | **Maximum score** | **4** |
| | **Positive cut-off value** | **≥2** | **Positive cut-off value** | **≥2** |

*GCS*: Glasgow Coma Scale; *qSOFA*: quick Sequential Organ Failure Assessment; *SIRS*: Systemic inflammatory response syndrome.

**Table 2. Components of NEWS, NEWS2, and MEWS.**

| Variable | NEWS | | NEWS2 | | MEWS | |
|---|---|---|---|---|---|---|
| | *Cut-off* | *Points* | *Cut-off* | *Points* | *Cut-off* | *Points* |
| AVPU | Alert | 0 | Alert | 0 | Alert | 0 |
| | VPU | 3 | CVPU* | 3 | React to voice (V) | 1 |
| | — | — | — | — | React to pain (P) | 2 |
| | — | — | — | — | Unresponsive (U) | 3 |
| HR (beats/min) | 51–90 | 0 | 51–90 | 0 | 51–100 | 0 |
| | 91–110; or 41–50 | 1 | 91–110; or 41–50 | 1 | 41–50 or 101–110 | 1 |
| | 111–130 | 2 | 111–130 | 2 | <40 or 111–129 | 2 |
| | ≤40 or ≥131 | 3 | ≤40 or ≥131 | 3 | ≥130 | 3 |
| $O_2$Sat (%) | ≥96 | 0 | ≥96† | 0 | — | — |
| | 94–95 | 1 | 94–95 | 1 | — | — |
| | 92–93 | 2 | 92–93 | 2 | — | — |
| | ≤91 | 3 | ≤91 | 3 | — | — |
| Oxygen supp. | No | 0 | No | 0 | — | — |
| | Yes | 2 | Yes | 2 | — | — |
| RR (breaths/min) | 12–20 | 0 | 12–20 | 0 | 9–14 | 0 |
| | 9–11 | 1 | 9–11 | 1 | 15–20 | 1 |
| | 21–24 | 2 | 21–24 | 2 | <9 or 21–29 | 2 |
| | ≤8 or ≥25 | 3 | ≤8 or ≥25 | 3 | ≥30 | 3 |
| SBP (mm Hg) | 111–219 | 0 | 111–219 | 0 | 101–199 | 0 |
| | 101–110 | 1 | 101–110 | 1 | 81–100 | 1 |
| | 91–100 | 2 | 91–100 | 2 | 71–80 or ≥200 | 2 |
| | ≤90 or ≥220 | 3 | ≤90 or ≥220 | 3 | ≤70 | 3 |
| Temperature (˚C) | 36.1–38 | 0 | 36.1–38 | 0 | 35–38.4 | 0 |
| | 35.1–36 or 38.1–39 | 1 | 35.1–36 or 38.1–39 | 1 | <35 or ≥38.5 | 2 |
| | ≥39.1 | 2 | ≥39.1 | 2 | — | — |
| | ≤35 | 3 | ≤35 | 3 | — | — |
| | **Maximum score** | 20 | **Maximum score** | 20 | **Maximum score** | 14 |
| | **Positive cut-off value** | ≥5 | **Positive cut-off value** | ≥5 | **Positive cut-off value** | ≥5 |

*AVPU*: Alert, verbal, pain, or unresponsive; *HR*: Heart rate; *NEWS*: National early warning score; *NEWS2*: National early warning score 2; *MEWS*: Modified early warning score; *$O_2$Sat*: Oxygen saturation; *RR*: Respiratory rate; *SBP*: Systolic blood pressure.

*—Level of consciousness and new confusion ('C'), thus AVPU becomes ACVPU, where C represents new confusion.

†—NEWS2 has a dedicated section (SpO$_2$ Scale 2) for use in patients with hypercapnic respiratory failure who have clinically recommended oxygen saturation of 88–92%.

predictive models have the same prognostic accuracy. Accordingly, we developed a protocol for a prognostic study to determine whether qSOFA has higher predictive performance for relevant clinical outcomes in adult patients with infection presenting to the ED. The primary outcome of this study is 30-days in-hospital mortality, and the secondary outcomes are ICU admission, prolonged ICU admission (i.e., ≥72 hours), 30- as well as 90-days all-cause mortality.

## Methods and analysis

### Study design and setting

This protocol describes a multicenter, prospective observational cohort study evaluating the prognostic accuracy of qSOFA score, SIRS criteria, and EWSs (NEWS/NEWS2/MEWS) for

in-hospital mortality among adult patients presenting to the ED with suspected infection (NCT05172479). The study's outlines are shown in Fig 1. The study duration is 12 months per center (9 months for recruitment and 3 months for follow-up). Recruiting centers and recruitment status are shown in Table 3 (July 2023).

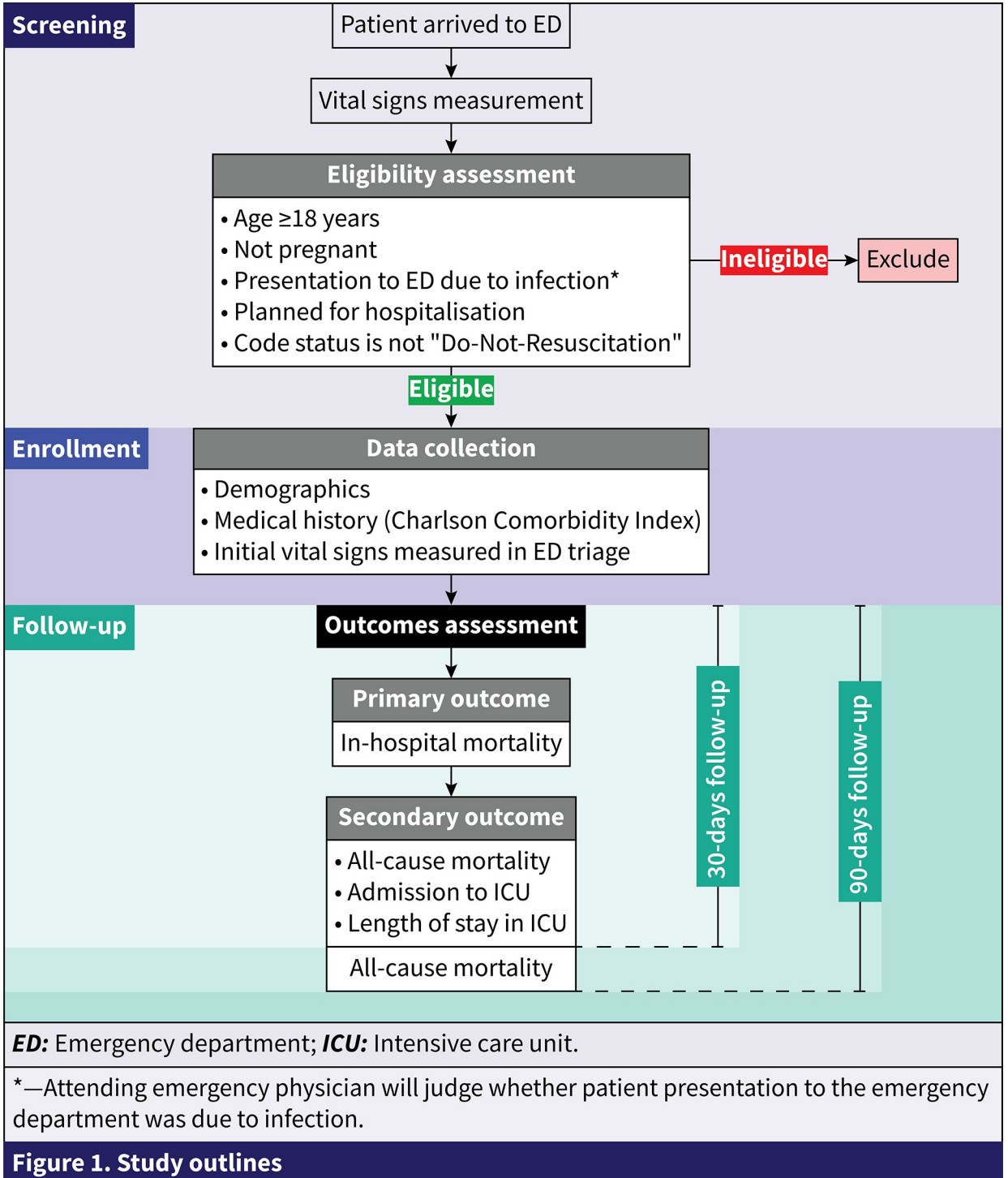

Fig 1. Study outlines. *—Attending emergency physician will judge whether patient presentation to the emergency department was due to infection. *ED*: Emergency department; *ICU*: Intensive care unit.

**Table 3. PASSEM study centers.**

| Country | City | Hospital | Status upon publication |
|---|---|---|---|
| Bahrain | Al Riffa | Bahrain Defence Force Hospital | Completed |
| | Muharraq | King Hamad University Hospital | Completed |
| Kuwait | Kuwait | Al-Amiri Hospital | Completed |
| Oman | Muscat | Armed Forces Hospital | Completed |
| Qatar | Doha | Hamad General Hospital | Completed |
| Saudi Arabia | Arar | North Medical Tower Hospital | Active, not recruiting |
| | Aseer Province | Aseer Central Hospital, Abha | Completed |
| | | Armed Forces Hospital Southern Region–Khamis Mushait | Active, not recruiting |
| | Eastern Province | Dr. Sulaiman Al Habib Hospitals–Khobar | Completed |
| | | Johns Hopkins Aramco Healthcare | Completed |
| | | King Fahad Specialist Hospital | Active, not recruiting |
| | | Royal Commission Hospital in Jubail | Completed |
| | Jeddah | King Abdulaziz University Hospital | Completed |
| | | King Fahad Armed Forces Hospital | Active, not recruiting |
| | Qassim Province | Dr. Sulaiman Al Habib Hospitals–Qassim | Completed |
| | Riyadh | Dr. Sulaiman Al Habib Hospitals–Riyadh | Completed |
| | | King Abdullah bin Abdulaziz University Hospital | Completed |
| | | King Fahad Medical City | Completed |
| | | King Khalid University Hospital | Completed |
| | | King Saud Medical City | Completed |
| Turkey | Kocaeli Province | Kocaeli University Hospital | Completed |
| United Arab Emirates | Abu Dhabi | Shaikh Shakhbout Medical City | Completed |
| | Dubai | Rashid Hospital | Completed |

## Diagnosis of infection

A presumptive diagnosis of infection will be judged based on the opinion of the ED physician upon the initial patient presentation. If required, two experts from each recruiting center will ascertain the diagnosis of infection on the 30[th] day since inclusion to the study. Evidence of infection will be sought by analyzing the patient's clinical, microbiological, and radiological data. Evidence of infection would be determined by either positive culture, other microbiological techniques (e.g., serological, or molecular), or radiological findings. If all of these evidence measures were equivocal, clinical context will be used to confirm the presence of infection. In cases of disagreement, consensus will be sought between the two experts. In all cases, the diagnosis of infection will be blinded to the output of the prediction models and the outcomes of patients.

## Study population

**Inclusion criteria.**  PSSEM study will enroll all consecutive adult patients (age ≥18 years) presenting to the ED with suspected infection who are planned for hospitalization (Box 1).

> Box 1. Eligibility criteria
>
> Inclusion criteria
>
> ▶ Adult patient (ages ≥18 years).

▶ Suspected infection (based on the opinion of the emergency physician).

▶ Planned for hospitalization.

Exclusion criteria

▶ Presentation to ED is not due to infection (e.g., autoimmune diseases, myocardial infarction, stroke, venous thromboembolism, trauma, intoxication . . . etc.).

▶ Pregnancy.

▶ Transferred from another hospitals.

▶ Code status is "Do-Not-Resuscitate" (DNR).

▶ Elective admission to the hospital (i.e., not through emergency department).

▶ Initial diagnosis of infection in the ED was not confirmed after finishing of the recruitment and follow-up phase.

**Exclusion criteria.**　We will exclude patients who present to the ED due to non-infectious causes (e.g., autoimmune diseases, myocardial infarction, trauma, . . .etc.), pregnant woman, those who are transferred from other hospitals, or with "Do-Not-Resuscitate" (DNR) code status. Patients whose initial diagnosis of infection in the ED was not confirmed after the recruitment and follow-up will also be excluded (Box 1).

## PASSEM study versus original derivation cohorts

The key characteristics of PASSEM Study and the original derivation cohorts of qSOFA, SIRS, and EWSs that will be assessed are shown in Table 4.

## Study flow chart

The study's procedures and assessments are shown in Table 5. Patients will undergo 4 phases: screening (Time$_{-1}$ [T$_{-1}$], 1–2 days), enrolment (T$_0$), and in-hospital (T$_1$, maximum 30 days after T$_0$), and out-hospital follow-up (T$_2$, maximum 90 days after T$_0$).

**Screening and enrolment phases.**　A staff member will screen patients for eligibility and check their measured vital signs once they arrive at the ED (triage) and the investigator will enroll potentially eligible patients (i.e., age ≥18 years, with suspected infection, and planned for hospitalization). First, a web-based electronic data capture system (EDC) will assign each patient to a participant number in ascending order. Then, the investigator will collect and enter the patient's initial data (demographics, contact information, Charlson Comorbidity Index (CCI) components, and variables required for qSOFA, SIRS, and EWS scores calculation) in an electronic case report form (eCRF) (see online supplementary materials). If the patient is not eligible, we will close the patient record in the EDC and clarify the cause of exclusion.

**In-hospital follow-up.**　Once enrolment is completed, the in-hospital follow-up phase will start (T$_1$, maximum 30 days after T$_0$) (Table 5). Study team will monitor hospitalized patients' status (i.e., death, alive and either discharged, transferred to another hospital, or still hospitalized) by consulting their specific medical registration number (MRN) in the recruiting center.

**Out-hospital follow-up.**　This phase starts if the patient is discharged from the hospital or 30-days have passed since inclusion to the study (whenever earlier; T$_2$, maximum 90 days after

**Table 4. Characteristics of PASSEM study and the original development cohorts of qSOFA score, SIRS criteria, NEWS/NEWS2, and MEWS.**

| Characteristic | PASSEM (n = 2,851) | qSOFA (n = 1,309,025) | SIRS (n = 519) | NEWS/NEWS2 (n = 35,585) | MEWS (n = 709) |
|---|---|---|---|---|---|
| Data collection period | 2021–2022 | 2010–2012 | 1992 | 2006–2008 | 2000 |
| Study design | Prospective cohort | Retrospective cohort | Prospective cohort | Retrospective cohort | Prospective cohort |
| Setting | 30 EDs across 7 countries | 12 community and academic hospitals in southwestern Pennsylvania (ED, hospital ward, and ICU) | 42 ICUs in 40 US hospitals | MAU at Portsmouth hospitals NHS Trust, UK | MAU at District General Hospital (DGH), UK |
| Definition of infection | Based on opinion of attending ED physician | Combination of body fluid culture and nonprophylactic antibiotic administration in the EHR | NA | NA | NA |
| Inclusion criteria | Adult patients (age ≥18 yrs.) with suspected infection who presented to the ED and planned for hospitalization | Adult patients (age ≥18 yrs.) with suspected infection | Patients with sepsis who lack a clear source of infection | All general medical emergency patients aged ≥16 yrs., except for those transferred directly to critical care areas of the hospital | All medical emergency admissions admitted to the MAU |
| Primary outcome | 30-days in-hospital mortality | In-hospital mortality | 24-hours in-hospital mortality | 24-hours in-hospital mortality | HDU or ICU admission, attendance of the cardiac arrest team at a cardiorespiratory emergency and death at 60 days |
| Time window for measuring variables | Initial presentation (at triage) | From 48 hrs. before to 24 hrs. after the onset of infection | Upon admission to ICU | NA | Twice daily for up to 5 days |

*ED*: Emergency department; *EHR*: Electronic health record; *HDU*: High dependency unit; *ICU*: Intensive care unit; *MAU*: Medical admission unit; *NA*: Not available; *NHS*: National Health Services; *UK*: United Kingdom; *US*: United States.

$T_0$) (Table 5). We will determine their status via telephone contact. We will also evaluate hospitalized patients' situations by consulting their MRN in the recruiting center. We will consider a patient lost to follow-up if we cannot reach them via telephone contact by the end of this phase.

**Table 5. Study's procedures and assessments.**

| Study components | Phases | | | |
|---|---|---|---|---|
| | Screening | Enrolment | In hospital F/U | Out hospital F/U |
| **Eligibility screening** | X | | | |
| **Data collection** | | | | |
| Demographics, medical history | | X | | |
| Physical examination/vital signs | | X | | |
| Blood investigations (WBCs count) | | X | | |
| **Primary outcome evaluation** | | | | |
| In-hospital mortality (within 30 days) | | | X | |
| **Secondary outcomes evaluation** | | | | |
| 1. ICU admission | | | X | |
| 2. ICU length of stay | | | X | |
| 3. All-cause mortality (within 30 days) | | | | X |
| 4. All-cause mortality (within 90 days) | | | | X |

*F/U*: Follow-up; *WBCs*: White blood cells

## Study outcome

The primary outcome of this study is 30-days in-hospital mortality. Secondary outcomes include ICU admission (within 30-days), ICU length of stay, and all-cause mortality within 30 and 90 days.

## Predictors

Lead investigator in each center will extract the demographics, components of CCI, vital signs, and blood investigations from the medical record of each potentially eligible patient. Study team will use the patient's initial vital signs, level of consciousness (i.e., first measurement in triage), WBC count, and partial pressure of carbon dioxide ($pCO_2$) to calculate qSOFA, SIRS, and EWSs. Blood pressure will be measured by using an electronic sphygmomanometer and results will be recorded in millimeters of mercury (mmHg). MAP will be calculated from SBP and diastolic blood pressure (DBP) using the following equation:

$$MAP = \frac{SBP + 2(DBP)}{3}$$

Pulse rate (recorded as beats/min) and oxygen saturation (recorded as a percentage) will be measured using an electronic pulse oximetry device. We will report whether the oxygen saturation reading was in room air or while a patient is on oxygen therapy. Body temperature will be measured orally or (axillary if necessary) by electronic thermometer and recorded as degree Celsius. A new-onset Glasgow Coma Scale (GCS) score of <15 will be considered significant for qSOFA calculation. If it is unclear whether a patient's confusion is 'new' or their usual state, we will assume the altered mental state/confusion is new until confirmed otherwise for all scores calculation.

Initial WBCs count (recorded in $x10^9/\mu L$) and $pCO_2$ (recorded in mmHg; if available) will be obtained from the patient's medical record and entered into the eCRF.

## Sample size

In the PASSEM study, we chose the method suggested by Collins et al [22]. In this method, sample size calculation is based on the expected event rate (minimum of 100 events in all validation datasets). However, rules-of-thumb for sample size are problematic, as they are not specific to the model or validation setting. Indeed, Snell et al showed that the rule-of-thumb of having at least 100 events and 100 non-events does not always produce precise estimates of a model's predictive performance measures [23]. To overcome this limitation, we chose to target an event rate of ≥200. Previous work by Freund et al [15] showed that a sample size of 879 patients yielded 74 events when power was set at 90%. Therefore, if we target a minimum of 200 events and consider 20% of lost to follow-up and missing data, a sample size of 2851 should be included. We will conduct an interim analysis after recruitment of 25%, 50%, and 75% of the target sample size to re-evaluate our assumptions and correct the sample size accordingly.

## Statistical analysis

Continuous data will be reported as mean (SD) or median (IQR) and compared using unpaired $t$ tests or analysis of variance and Mann-Whitney or Kruskal-Wallis test. Categorical variables will be expressed as number (percentage) and compared using a $\chi^2$ test or a Fisher exact test. We will begin by calculating an overall area under the receiver operating characteristic curve (AUC of ROC curve) and generate calibration curves of the qSOFA, SIRS, and EWSs

to predict the primary and secondary outcomes. Subsequent to assessing the model's overall performance; sensitivity, specificity, positive and negative predictive values will be calculated with cross tables for predicting primary and secondary outcomes for a qSOFA score of $\geq 2$, SIRS of $\geq 2$, and EWSs of $\geq 5$. We will use the Kaplan-Meier method to estimate in-hospital and 90-day all-cause mortality. A log-rank regressions will be used to assess groups' differences. Odd ratios (ORs) for in-hospital death, ICU admission, and 90-days all-cause mortality of qSOFA, SIRS, and EWSs will be estimated with a logistic regression analysis after adjustment for the patients' demographics, comorbidities, and CCI. The model fit will be assessed by the calculation of the log-likelihood, Akaike information criterion (AIC), AUC, Bayesian information criterion (BIC), and D-statistics. To compare the performance of qSOFA, SIRS and EWSs, we will use absolute net-reclassification index (NRI). The absolute NRI mathematically represents a net proportion of patients correctly reclassified by one model as compared to another [24]. Net reclassification involves classifying patients in risk categories and determining how a new model reclassifies patients into various risk categories compared with a previous model. Risk differences are classified based on the actual outcome patients experienced (those who died vs those who did not).

A priori subgroup analyses will be conducted based on status of the following: COVID-19 (present vs absent), febrile neutropenia (present vs absent), solid organs or hematological cancers (present vs absent), autoimmune diseases (present vs absent), and severe comorbidities (CCI $\geq 3$ vs $<3$), and race of the patient (Asian vs Black vs South Asian vs White) as permitted by sample size. If missing data is minimal ($<5\%$) we will conduct a complete case analysis, otherwise we will use multiple imputation.

For all analyses, a 2-tailed $P < 0.05$ will be considered statistically significant. Statistical analyses will be performed with Stata version 17.0 [25] and RStudio version '2022.7.0.548' [26].

## Data management

We will use an encrypted, web-based EDC (Castor®) for this study [27]. Lead investigators (or their delegates) will enter clinical data on an eCRF at each participating center. They will make all entries, corrections, and alterations. The data manager of this study will provide all tools, instructions, and training necessary to complete the eCRF, and each user will be issued a unique username and password.

The monitors will review the eCRFs, evaluate them for completeness and consistency, and compare them with the source documents to ensure no discrepancies. The Monitors cannot enter data in the eCRFs. Lead investigators must verify that all data entries in the eCRF are accurate and correct. If some assessments are not done, or specific information is unavailable, not applicable, or unknown, they must indicate this in the eCRF. Finally, lead investigators must electronically sign off all patients' eCRF enrolled from their hospitals.

Data manager will lock the final validated database so that no more change will be possible on the frozen data. Subsequently, the principal investigator will receive the patient data (eCRF data + audit trail) for archiving at the investigational site and transference in a secure way to the biostatistical team in Stata format.

## Ethics

**Informed consent.** Informed consent was waived for this study due to its complete observational nature and absence of interventions or invasive procedures. The study does not impose any change in the standard practice of sepsis at the site; and the patient's data will be collected prospectively from their medical record at recruiting centers. The benefit/risk ratio of participation in the study is excellent. Moreover, we expect that PASSEM Study results may

improve patient care in the recruiting center by allowing a better understanding of ideal tools to identify patients with sepsis.

**Ethical approval.** This protocol complies with the principles laid down by the 59th World Medical Assembly and all applicable amendments laid down by the World Medical Assemblies, the applicable regulations per site, and any other relevant local requirement and laws.

PASSEM study has been approved by local Institution review board (LIRB) of all recruiting hospitals at the time of publication of this protocol. Data manager of PASSEM study will not grant access to the EDC system or start the study until the principal investigator receives a copy of a written and dated approval/favorable signed opinion from each participating center LIRB.

We will present any change in this protocol as an amendment in written form to the protocol. The principal investigator and lead investigators will sign the protocol amendment and then submitted to the LIRBs. Following approval, we will send the amendment to all participating investigators. The amendment cannot be acted upon before the outcome of this decision. However, the study team will submit minor modifications (administrative modifications, including a new recruitment center) to the LIRBs for information purposes only.

## Patient confidentiality

In order to maintain confidentiality, we will not collect any patient's-identifying data (e.g., name, identification number, medical record number [MRN], etc.) in the eCRF. Instead, lead investigators will store such data in a separate list sheet specified for each participating center. The lead investigator of each center will maintain this list in strict confidence.

## Discussion

PASSEM study will be the first international multicenter prospective cohort study that designated to externally validate qSOFA score, SIRS criteria, and EWSs for in-hospital mortality among adult patients with suspected infection presenting to the ED in the Middle East region. In an ED setting, it is crucial to take a patient's vital signs as early as possible to make decisions and predict the patient's outcome. Hence, PASSEM study will use initial physiologic parameters the patient presented with, to the ED (triage vital signs) to calculate qSOFA, SIRS criteria, and EWSs. Furthermore, definition of infection will be based on the opinion of the ED attending physician with subsequent confirmation at the end of in-hospital follow-up, which might be more appropriate and pragmatic.

We will publish study's results in peer-reviewed journals and may present them at scientific conferences. We will follow recommendations of Transparent Reporting of a Multivariable Prediction Model For Individual Prognosis or Diagnosis (TRIPOD) guidelines [28]. The most significant results will be shared to the public through social networks.

## Supporting information

**S1 File.**
(PDF)

## Acknowledgments

The authors would like to thank professors and doctors Gordon H. Guyatt (McMaster University, Canada), Yonathan Freund (Sorbonne University, France), David Pilcher (Alfred Hospital, Australia), and Mohammed Alshahrani (King Fahd University Hospital, Saudi Arabia) for their comments, feedback, and advice.

## Author Contributions

**Conceptualization:** Abdullah M. Algarni, Musa S. Alfaifi, Ali A. Al Bshabshe, Othman M. Omair, Mohammed A. Alsultan, Hasan M. Alzahrani, Hadi E. Alali, Abdullah A. Alsabaani, Farid Foroutan.

**Data curation:** Abdullah M. Algarni, Ali A. Al Bshabshe, Abdullah A. Alsabaani, Ali M. Alqarni, Ahmed A. Alsaad, Naif T. Marzook, Abdullah A. Basri, Abdallah M. Elsafti, Abdulaziz S. Alalshaikh, Cansu A. Özturan, Yousef I. Alawad, Awad AlOmari, Malek A. Alkhateeb, Moonis M. Farooq, Latifa Ali AlMutairi, Muneera M. Alasfour, Mohammad I. Al Haber, Umma-Kulthum A. Umar, Nidal H. Bokhary, Saeed F. Alqahtani, Abdulrhman Almutairi, Hisham F. Alyahya, Wejdan S. Alzahrani, Fawziah Alsalmi, Abdulmajeed M. Omair, Faysal M. Alasmari, Sultan Y. Alfifi, Mohammed S. Al-Nujimi, Farid Foroutan.

**Formal analysis:** Abdullah M. Algarni, Abdullah A. Alsabaani, Farid Foroutan.

**Methodology:** Abdullah M. Algarni, Musa S. Alfaifi, Ali A. Al Bshabshe, Mohammed A. Alsultan, Abdullah A. Alsabaani, Farid Foroutan.

**Project administration:** Abdullah M. Algarni, Musa S. Alfaifi, Ali A. Al Bshabshe, Othman M. Omair, Hadi E. Alali, Ali M. Alqarni, Salah A. Alghanem, Bandar S. Al Mufareh, Ayesha M. Almemari, Abdulrahman A. Sindi, Ibrahim U. Ozturan, Abdullah A. Alhadhira, Asaad S. Shujaa, Abdullah H. Alotaibi, Moosa M. Awladthani, Abdullah A. Almarshed, AlHanouv M. AlQahtani, Tim R. Harris, Bader A. Alyahya, Saad A. Assiri, Feras H. Abuzeyad, Sara N. Kazim, Abdulrahman A. Al-Fares, Faisal Y. Almazroua, Naif T. Marzook, Abdallah M. Elsafti, Cansu A. Özturan, Awad AlOmari, Malek A. Alkhateeb, Latifa Ali AlMutairi, Mohammad I. Al Haber, Umma-Kulthum A. Umar, Abdulrhman Almutairi, Hisham F. Alyahya.

**Resources:** Abdullah M. Algarni, Musa S. Alfaifi, Tim R. Harris, Saeed F. Alqahtani.

**Software:** Abdullah M. Algarni, Ali M. Alqarni, Farid Foroutan.

**Supervision:** Abdullah M. Algarni, Musa S. Alfaifi, Ali A. Al Bshabshe, Othman M. Omair, Hadi E. Alali, Abdullah A. Alsabaani, Ali M. Alqarni, Salah A. Alghanem, Bandar S. Al Mufareh, Ayesha M. Almemari, Abdulrahman A. Sindi, Ibrahim U. Ozturan, Abdullah A. Alhadhira, Asaad S. Shujaa, Abdullah H. Alotaibi, Moosa M. Awladthani, Abdullah A. Almarshed, AlHanouv M. AlQahtani, Tim R. Harris, Bader A. Alyahya, Saad A. Assiri, Feras H. Abuzeyad, Sara N. Kazim, Abdulrahman A. Al-Fares, Faisal Y. Almazroua, Naif T. Marzook, Abdallah M. Elsafti, Cansu A. Özturan, Awad AlOmari, Malek A. Alkhateeb, Latifa Ali AlMutairi, Mohammad I. Al Haber, Umma-Kulthum A. Umar, Saeed F. Alqahtani, Abdulrhman Almutairi, Hisham F. Alyahya.

**Validation:** Abdullah M. Algarni, Ali A. Al Bshabshe, Othman M. Omair, Hadi E. Alali, Ali M. Alqarni, Salah A. Alghanem, Bandar S. Al Mufareh, Ayesha M. Almemari, Abdulrahman A. Sindi, Ibrahim U. Ozturan, Abdullah A. Alhadhira, Asaad S. Shujaa, Abdullah H. Alotaibi, Moosa M. Awladthani, Abdullah A. Almarshed, AlHanouv M. AlQahtani, Bader A. Alyahya, Saad A. Assiri, Feras H. Abuzeyad, Sara N. Kazim, Abdulrahman A. Al-Fares, Faisal Y. Almazroua, Naif T. Marzook, Cansu A. Özturan, Awad AlOmari, Malek A. Alkhateeb, Latifa Ali AlMutairi, Mohammad I. Al Haber, Umma-Kulthum A. Umar, Saeed F. Alqahtani, Abdulrhman Almutairi, Farid Foroutan.

**Visualization:** Abdullah M. Algarni, Abdullah A. Alsabaani, Hisham F. Alyahya, Farid Foroutan.

**Writing – original draft:** Abdullah M. Algarni, Ali A. Al Bshabshe, Mohammed A. Alsultan, Hasan M. Alzahrani, Abdullah A. Alsabaani, Ali M. Alqarni, Tim R. Harris, Farid Foroutan.

**Writing – review & editing:** Abdullah M. Algarni, Musa S. Alfaifi, Ali A. Al Bshabshe, Othman M. Omair, Mohammed A. Alsultan, Hasan M. Alzahrani, Hadi E. Alali, Abdullah A. Alsabaani, Ali M. Alqarni, Salah A. Alghanem, Bandar S. Al Mufareh, Ayesha M. Almemari, Abdulrahman A. Sindi, Ibrahim U. Ozturan, Abdullah A. Alhadhira, Asaad S. Shujaa, Abdullah H. Alotaibi, Moosa M. Awladthani, Ahmed A. Alsaad, Abdullah A. Almarshed, AlHanouv M. AlQahtani, Tim R. Harris, Bader A. Alyahya, Saad A. Assiri, Feras H. Abuzeyad, Sara N. Kazim, Abdulrahman A. Al-Fares, Faisal Y. Almazroua, Naif T. Marzook, Abdullah A. Basri, Abdallah M. Elsafti, Abdulaziz S. Alalshaikh, Cansu A. Özturan, Yousef I. Alawad, Awad AlOmari, Malek A. Alkhateeb, Moonis M. Farooq, Latifa Ali AlMutairi, Muneera M. Alasfour, Mohammad I. Al Haber, Umma-Kulthum A. Umar, Nidal H. Bokhary, Saeed F. Alqahtani, Abdulrhman Almutairi, Hisham F. Alyahya, Wejdan S. Alzahrani, Fawziah Alsalmi, Abdulmajeed M. Omair, Faysal M. Alasmari, Sultan Y. Alfifi, Mohammed S. Al-Nujimi, Farid Foroutan.

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
