## [Decision Letter · Decision Letter 0]

12 Aug 2022

PONE-D-22-10909Prognostic accuracy of qSOFA score, SIRS criteria, and EWSs for in-hospital mortality among adult patients presenting with suspected infection to the emergency department (PASSEM): Multicenter prospective external validation cohort study protocol

PLOS ONE

Dear Dr. Algarni,

Thank you for submitting your manuscript to PLOS ONE. After careful consideration, we feel that it has merit but does not fully meet PLOS ONE’s publication criteria as it currently stands. Therefore, we invite you to submit a revised version of the manuscript that addresses the points raised during the review process.

The manuscript has been evaluated by one reviewer, and his comments are available below.

The reviewer has raised a number of concerns. He requests improvements to the reporting of methodological aspects of the study, for example, regarding the primary end-point and the partial pressure of carbon dioxide.  

Could you please carefully revise the manuscript to address all comments raised?

A marked-up copy of your manuscript that highlights changes made to the original version. You should upload this as a separate file labeled 'Revised Manuscript with Track Changes'.An unmarked version of your revised paper without tracked changes. You should upload this as a separate file labeled 'Manuscript'.

We look forward to receiving your revised manuscript.

Kind regards,

Lorena Verduci

Staff Editor

PLOS ONE

Journal Requirements:

Reviewers' comments:

Reviewer's Responses to Questions

**Comments to the Author**

1. Does the manuscript provide a valid rationale for the proposed study, with clearly identified and justified research questions?

Reviewer #1: Partly

2. Is the protocol technically sound and planned in a manner that will lead to a meaningful outcome and allow testing the stated hypotheses?

Reviewer #1: Yes

3. Is the methodology feasible and described in sufficient detail to allow the work to be replicable?

Reviewer #1: Yes

4. Have the authors described where all data underlying the findings will be made available when the study is complete?

Reviewer #1: Yes

5. Is the manuscript presented in an intelligible fashion and written in standard English?

Reviewer #1: Yes

6. Review Comments to the Author

You may also provide optional suggestions and comments to authors that they might find helpful in planning their study.

Reviewer #1: Thank you for the opportunity to review this study protocol entitled, ‘Prognostic accuracy of qSOFA score, SIRS criteria and EWSs for in-hospital mortality among adult patients presenting with suspected infection to the emergency department (PASSEM): Multicenter prospective external validation cohort study protocol’. In this prospective multicentre observational study, the authors plan to validate multiple scores to study their prognostic accuracy for 30 day mortality in patients with an infection. The protocol is well written and reasonably easy to follow. The authors should be commended for undertaking such a large multicentre study. However, I do have some minor concerns. They are as follows:

1) Primary end-point: The abstract states ‘30 day in-hospital mortality’. Not all patients are admitted for 30 days. So, I would expect the primary end-point to be either 30 day mortality or in-hospital mortality. Please clarify the primary end-point.

2) Primary and secondary end-points: Please sate the primary and secondary end points at the end of the introduction.

3) Hypothesis: The authors have hypothesized that qSOFA has greater prognostic accuracy. It is usual to have a null-hypothesis and the result should either maintain or reject the null-hypothesis. Could this be reworded as such?

4) Partial pressure of carbon dioxide: The authors have stated that they will be noting the partial pressure of carbon dioxide. Does this mean all patients will have an arterial blood gas? What is the purpose? Is it to determine if the patient is a CO2 retainer? If so, is this parameter only for those with COPD?

5) Follow-up: The authors have stated that ‘the study does not impose any change in the standard practice’. The protocol also states, ‘we will determine their status via telephone contact’. Is it standard practice to telephone patients to determine their status? Do patient not need to consent to receiving a telephone call?

7. PLOS authors have the option to publish the peer review history of their article (what does this mean?). If published, this will include your full peer review and any attached files.

Reviewer #1: **Yes: **Dr Narani Sivayoham

---

## [Author Response · Author response to Decision Letter 0]

9 Sep 2022

Dear Editors,

Thank you for the opportunity to address the comments from the Dr. Narani Sivayoham. The authors hope that the Reviewer and Editors will be satisfied with the further amendments which we have made to the manuscript.

We believe that the manuscript is now suitable for publication in PLOS ONE.

Sincerely regards,

Abdullah M. Algarni, MBBS

Family medicine consultant

Aseer Central Hospital

Abha, Saudi Arabia

---

## [Decision Letter · Decision Letter 1]

18 Jan 2023

PONE-D-22-10909R1

Prognostic accuracy of qSOFA score, SIRS criteria, and EWSs for in-hospital mortality among adult patients presenting with suspected infection to the emergency department (PASSEM): Multicenter prospective external validation cohort study protocol

PLOS ONE

Dear Dr. Abdullah, 

We’re pleased to inform you that your manuscript has been judged scientifically suitable for publication and will be formally accepted for publication once it meets all outstanding technical requirements.

Kind regards,

Yaser Mohammed Al-Worafi

Academic Editor

PLOS ONE

Additional comments: 

Reviewer 2's concerns appear to be outside the scope of Study Protocols (which are designed only to report a plan for a future or ongoing study, not to report new data), so we support an Accept decision.

---

## [Editor Report · Acceptance letter]

10 Jul 2023

PONE-D-22-10909R1 

Prognostic accuracy of qSOFA score, SIRS criteria, and EWSs for in-hospital mortality among adult patients presenting with suspected infection to the emergency department (PASSEM)
Multicenter prospective external validation cohort study protocol 

Dear Dr. Algarni:

I'm pleased to inform you that your manuscript has been deemed suitable for publication in PLOS ONE. Congratulations! Your manuscript is now with our production department. 

Kind regards, 

on behalf of

Professor Yaser Mohammed Al-Worafi 

Academic Editor

PLOS ONE